# The Multifaceted Presentation of the Multisystem Inflammatory Syndrome in Children: Data from a Cluster Analysis

**DOI:** 10.3390/jcm11061742

**Published:** 2022-03-21

**Authors:** Hafize Emine Sönmez, Şengül Çağlayan, Gülçin Otar Yener, Eviç Zeynep Başar, Kadir Ulu, Mustafa Çakan, Vafa Guliyeva, Esra Bağlan, Kübra Öztürk, Demet Demirkol, Ferhat Demir, Şerife Gül Karadağ, Semanur Özdel, Nuray Aktay Ayaz, Betül Sözeri

**Affiliations:** 1Department of Pediatric Rheumatology, Kocaeli University, Kocaeli 41001, Turkey; eminesonmez@gmail.com; 2Department of Pediatric Rheumatology, Umraniye Training and Research Hospital, University of Health Sciences, Istanbul 34766, Turkey; sengulturkercaglayan@gmail.com (Ş.Ç.); drkadirulu@gmail.com (K.U.); mustafacakan@hotmail.com (M.Ç.); drferhat@outlook.com (F.D.); 3Department of Pediatric Rheumatology, Şanlıurfa Research and Training Hospital, Şanlıurfa 63330, Turkey; gulcinotar@gmail.com; 4Department of Pediatric Cardiology, Kocaeli University, Kocaeli 41001, Turkey; evicbasar@gmail.com; 5Department of Pediatric Rheumatology, Faculty of Medicine, Istanbul University, Istanbul 34093, Turkey; doktor_guliyeva@hotmail.com (V.G.); nurayaktay@gmail.com (N.A.A.); 6Department of Pediatric Rheumatology, Sami Ulus Research and Training Hospital, Ankara 06560, Turkey; eozb@yahoo.com (E.B.); semanurozdel@gmail.com (S.Ö.); 7Department of Pediatric Rheumatology, Göztepe Research and Training Hospital, Istanbul Medeniyet University, Istanbul 34722, Turkey; ozturk1209@gmail.com; 8Department of Pediatric Intensive Care, Faculty of Medicine, Istanbul University, Istanbul 34093, Turkey; ddemirkol@istanbul.edu.tr; 9Department of Pediatric Rheumatology, Erzurum Regional Training and Research Hospital, Erzurum 25070, Turkey; sgulkaradag@gmail.com

**Keywords:** multisystem inflammatory syndrome in children (MIS-C), COVID-19, cluster analysis

## Abstract

Background: The aim of this study was to evaluate the outcomes of patients with the multisystem inflammatory syndrome in children (MIS-C) according to phenotypes of disease and define the prognostic factors for the severe course. Methods: This cross-sectional study included 293 patients with MIS-C from seven pediatric rheumatology centers. A two-step cluster analysis was performed to define the spectrum of disease and their outcomes were compared between each group. Results: Four subgroups were identified as follows: cluster I, predominantly Kawasaki-like features (*n* = 100); cluster II, predominantly MAS-like features (*n* = 34); cluster III, predominantly LV dysfunction (*n* = 47); cluster IV, other presentations (*n* = 112). The duration of fever was longer in cluster II and the length of hospitalization was longer in both clusters II and III. Laboratory findings revealed lower lymphocyte and platelet counts and higher acute phase reactants (APRs) in cluster II, while patients in cluster IV showed less inflammation with lower APRs. The resolution of abnormal laboratory findings was longer in clusters II and III, while it was shortest in cluster IV. Seven patients died. Among them, four belonged to cluster II, while three were labeled as cluster III. Patients with severe course had higher levels of neutrophil–lymphocyte ratio, mean platelet volume, procalcitonin, ferritin, interleukin-6, fibrinogen, D-Dimer, BNP, and troponin-I, and lower levels of lymphocyte and platelet counts. Conclusion: As shown, MIS-C is not a single disease presenting with various clinical features and outcomes. Understanding the disease spectrum will provide individualized management.

## 1. Introduction

Multisystem inflammatory syndrome in Children (MIS-C) is a recently defined disorder with a dysregulated immune response in etiopathology. Patients are mainly characterized by fever, rash, and systemic inflammation, resulting in multisystem organ dysfunction around 4–5 weeks after SARS-CoV-2 infection [1,2]. Initially, the US Centers for Disease Control and Prevention (CDC) announced the characteristics of disease as follows: age of younger than 21 years; fever (body temperature, ≥38.0 °C) lasting at least 24 h; serious illness leading to hospitalization; laboratory evidence of inflammation; multisystem organ involvement involving at least two systems; laboratory-confirmed SARS-CoV-2 infection (positive SARS-CoV-2 real-time reverse transcriptase–polymerase chain (RT–PCR) or antibody test during hospitalization) or a contact to a person with suspected or confirmed COVID-19 within 4 weeks before the onset of symptoms [2]. These criteria were introduced by employing limited data extracted from early reported cases. However, it is now well known that the disease does not have a single phenotype and that various symptoms and system involvements may accompany the disease [3]. Furthermore, patients presenting several months after a documented COVID-19 illness or after contact with a person having COVID-19 were reported [4]. The spectrum of the disease has become a subject of increasing interest, as patients with MIS-C express a phenotype similar to many diseases, such as Kawasaki disease (KD), macrophage-activating syndrome (MAS), and septic shock. Most recently, Geva et al. [5] performed a data-driven cluster analysis (CA) to identify the subphenotypes of SARS-CoV-2 during childhood and showed that three clusters of disease were as follows: cluster 1 (previously healthy individuals with a mean age of 7.2 years, presenting with mainly cardiovascular and/or mucocutaneous features and negative for nasopharynx SARS-CoV-2 PCR); cluster 2 (individuals with frequent respiratory findings and positive PCR test); cluster 3 (individuals with younger age (mean 2.8), positive PCR test and less inflammation). According to their data, patients with pulmonary findings and positive SARS-CoV-2 PCR test were labeled as cluster 2, while MIS-C patients were distinguished as a separate subgroup (cluster 1) [5]. However, data on clusters of MIS-C patients and outcomes according to clusters have not been examined meticulously. Cluster analysis is a different statistical method that provides clinicians to reveal hidden phenotypes of diseases in large datasets. The goal of CA is to distinguish subphenotypes of disease by clustering them into comparable groups. With this method, researchers may reach relatively homogeneous subgroups and may individualize management or have an opportunity to study etiopathogenesis in a more homogeneous group.

Herein, the primary aim of the study is to define the spectrum and phenotypic characteristics and outcomes of MIS-C patients by using the CA method, and the secondary aim is to display the prognostic factors for the severe course.

## 2. Methods

This cross-sectional study was performed between May 2020 and December 2021. Patients’ demographic data, clinical and laboratory features, treatments, and outcomes were extracted from the Pediatric Rheumatology Academy (PeRA)-Research Group (RG) database [6]. PeRA-RG is a newly formed group collecting data for prospective observational cohort studies consisting of pediatric rheumatologists from seven pediatric rheumatology centers. Each center has entered the data of their patients with MIS-C since May 2020. All of the data were collected anonymously. Patients who met the WHO or CDC criteria [1,2] and were treated according to American College of Rheumatology (ACR) Clinical Guidance [7] were included to minimize center-related inconstancy. Previously reported cases were included with appropriate references.

A severe case was defined as the presence of a requirement for O_2_ support, vasoactive agents, or having multiple organ dysfunction syndrome (MODS). MODS is defined as the progressive physiological dysfunction of two or more organ systems where homeostasis cannot be maintained without intervention.

Definition of complete and incomplete KD was made on American Heart Association (AHA) definition [8]. The diagnoses of MAS were made according to the MAS classification criteria [9].

Less than the 5th percentile of age and gender-specific blood pressure levels were defined as hypotension. Heart rates were evaluated according to the normal range for age. Left ventricular dysfunction was defined as decreased EF (<55%) or FS (<28%). Coronary artery abnormalities were defined according to the AHA guidelines for KD as dilation (z-score = 2.0–2.49), small aneurysm (z-score = 2.5–4.9), moderate aneurysm (z-score = 5–9.9), and large/giant aneurysm (z-score ≥ 10) [8]. Mitral valve insufficiency was determined as the malfunction of the mitral valve that results in regurgitation from the left ventricle into the left atrium.

Patients with a concomitant rheumatic disease were excluded.

Ethical approval was obtained from the local ethics committee (Umraniye Training and Research Hospital, University of Health Sciences, Istanbul. Approval number: B.10.1.TKH.4.34.H.GP.0.01/27).

### Statistical Analyses

The SPSS version 21.0 (SPSS, Inc., Chicago, IL, USA) was used for statistical analysis. The variables were investigated using visual (histogram, probability plots) and analytic methods (Kolmogorov–Smirnov/Shapiro–Wilk’s test) to determine whether or not they were normally distributed. Descriptive analysis was presented using proportions, mean, standard deviation (SD), medians, minimum (min), and maximum (max) values as appropriate. Categorical data were statistically analyzed by Fisher’s exact or chi-square tests, as appropriate. Continuous data were analyzed by Student’s *t*-test or Mann–Whitney U test, whichever was appropriate. *p* values < 0.05 were considered statistically significant. Cluster analysis was performed by using the two-step cluster analysis method. Initially, variables were selected such as continuous or categorical. The current age, duration of symptoms, and laboratory parameters were the continuous variables, while gender and clinical manifestations were the categorical variables.

## 3. Results

### 3.1. The Clinical and Laboratory Findings of All Patients

A total of 293 patients with MIS-C were evaluated. Among them, 176 (60.1%) were male, and 117 (39.9%) were female. The median (min–max) age of patients was 88 (1–214) months. In total, 30 (10.2%) patients were obese (age and gender-specific body mass index [BMI] ≥ 95th percentile), and 14 (4.7%) had a concomitant disease (neurologic disease = 6, endocrinologic disease = 2, cardiologic disease = 2, asthma = 2, hydronephrosis = 1, and thalassemia minor = 1).

Of 293 patients, 217 (74.1%) had a history of close contact with a symptomatic COVID-19 patient. Nasopharynx SARS-CoV-2 PCR was positive in 12 (4.1%) patients. The results of the SARS-CoV-2 antibody test were available for all the patients. Of 293 patients, 282 (96.2%) had positive results of SARS-CoV-2 IgG and/or IgM antibodies.

All patients suffered from fever lasting a median duration of 5 (1–30) days. The most common clinical features were as follows: gastrointestinal involvement in 197 (67.2%), mucocutaneous features in 173 (59%), and cardiac involvement in 90 (30.7%) patients. The baseline clinical characteristics of MIS-C patients are depicted in Table 1.

At admission, laboratory examination revealed lymphopenia (<1500 mm^3^) in 183 (62.4%) and thrombocytopenia (<150,000 mm^3^) in 80 (27.3%) patients. Elevated CRP levels were detected in 265 (90.4%), ESR was elevated in 183 (62.4%), ferritin was increased in 163 (55.6%), and procalcitonin was increased in 193 (65.8%) patients. A total of 10 (3.4%) patients had elevated troponin-I levels, while an increased level of pro-BNP was detected in 143 (48.8%) patients. D-dimer was increased in 221 (75.4%) patients. The Laboratory findings of patients at admission are depicted in Table 2.

### 3.2. Cluster Analysis of Patients

According to the two-step cluster analysis, four distinct subgroups were identified: cluster I, with predominantly Kawasaki-like features (*n* = 100); cluster II, with predominantly MAS-like features (*n* = 34); cluster III, with predominantly LV dysfunction (*n* = 47); cluster IV, comprising other presentations (*n* = 112). Patients in cluster I were younger than patients from other groups, while no gender differences were observed. The duration of fever was longer in cluster II, and the length of hospitalization was longer in both clusters II and III. Conjunctivitis, polymorphous rash, oral changes, extremity changes, and cervical lymphadenopathy were more common in cluster I. Respiratory symptoms were more common in clusters II and III, while there were no differences between clusters in terms of neurologic, renal, and gastrointestinal involvements. Left ventricular (LV) systolic dysfunction was more frequent in cluster III, whereas coronary artery involvement was more common in cluster I. Laboratory findings showed lower lymphocyte and platelet counts and higher CRP, ferritin, procalcitonin levels in cluster II while patients in cluster IV showed less inflammation with lower CRP, ferritin, IL-6, and procalcitonin levels. The pro-BNP and troponin-I levels were higher in cluster III, as expected (Table 3).

### 3.3. Treatments and Outcomes of Patients According to Clusters

The median time of hospitalization was 11 (3–39). A total of 87 (29.6%) patients required intensive care after a median of 1 day of hospitalization. The median length of stay in intensive care was 5 (2–16) days.

In total, 272 (92.8%) patients received intravenous immunoglobulin (IVIG) (2 g/kg), and 33 (11.3%) patients required a second dose of IVIG. Furthermore, 53 (18.1%) patients were additionally treated with pulse methylprednisolone (10–30 mg/kg/day, 1–3 days), and 180 (61.4%) were also treated with methylprednisolone (2 mg/kg/day). Steroid treatment was tapered and ceased within a median of 20 days. In addition, 2 (0.7%) patients were treated with tocilizumab, and 77 (26.3%) patients received anakinra, concomitantly. Therapeutic plasma exchange was applied to 24 (8.1%) patients. In total, 52 (17.7%) patients required inotropic agents. Furthermore, 236 (80.5%) patients were treated with prophylactic low-molecular-weight heparin (LMWH) (1 mg/kg/day), and 49 (16.7%) were treated with acetylsalicylic acid (ASA). The need for intensive care, pulse steroid, anakinra, and inotropic agents were more common in both clusters II and III.

Fever was the first symptom that resolved with a median of 6 days after initiation of treatment, and the lymphocyte count was the first laboratory marker that returned to the normal range. The lymphocyte count increased to over 1500 mm^3^ within a median of 13 days. CRP, D-dimer, and BNP levels reached normal levels within a median of 14, 20, and 32 days, respectively. EF returned to normal within 11.5 days, while during the follow-up, abnormal echocardiographic finding continued in 6 patients (mild mitral valve insufficiency = 3, pericardial effusion = 1, coronary artery dilatation = 2). The recovery duration of abnormal laboratory findings was longer in clusters II and III, while it was shortest in clusters IV. Seven patients died. Among them, four belonged to cluster II, and three were labeled as cluster III.

### 3.4. The Comparison of Severe Cases and Others

In total, 81 patients were classified as severe. When the patients with the severe course were compared with the others, they were older and required longer hospitalization. Splenomegaly, hepatomegaly, respiratory findings, cardiac, renal, and neurologic involvements were more common in patients with the severe course. Laboratory evaluation of patients with severe course revealed higher levels of neutrophil–lymphocyte ratio (NLR), mean platelet volume (MPV), procalcitonin, ferritin, interleukin (IL)-6, fibrinogen, D-Dimer, pro-BNP, and troponin-I, and lower levels of lymphocyte and platelet counts (Table 4).

## 4. Discussion

Herein, we showed that MIS-C is not a single disease presenting with various symptoms and usually clustering as four distinct groups. Besides the clinical and laboratory features, there are certain differences between clusters in terms of outcomes and treatment requirements as well. Recent data from the USA resulting from an evaluation of the characteristics of hospitalized children with COVID-19 revealed that 2.7% of hospitalized children with COVID-19 presented with MIS-C [10]. Furthermore, Verdoni et al. [11] reported a 30-fold increased risk for that Kawasaki-like disease in patients infected with SARS-CoV-2, and in the same line, they declared an increased risk of MAS and shock in patients with MIS-C. MIS-C is a rare and serious inflammatory condition associated with COVID-19. The mortality rate varies from 1% to 9% according to multicenter studies [12,13,14]. Although initial reports announced this new condition as a Kawasaki-like disease, it is well known now that patients with MIS-C may present with different system involvements. A recent meta-analysis showed the pooled estimated rate of most common clinical features as follows: fever (proportion = 82.4%, 95% CI 69.8–95.1%); polymorphous rash (proportion = 63.7%, 95% CI 53.8–73.5%); gastrointestinal symptoms (proportion = 79.4%, 95% CI 68.1–90.7%); shock (proportion = 68.1%, 95% CI 51.9–84.3%); neurocognitive symptoms (proportion = 31.8%, 95% CI 21.9–41.8%); acute kidney injury (proportion = 41.10%, 95% CI 15.0, 67.1) [15]. The spectrum of the disease may vary in each patient. Some patients may present with persistent fever with milder symptoms in the absence of severe systemic involvement, while some MIS-C patients deteriorate abruptly and require PICU [16]. By phenotyping clusters for clarifying the disease spectrum, the management of the disease may be individualized. A latent class analysis from the USA showed three classes of MIS-C patients with different manifestations [16]. In the aforementioned study, 35.6% of patients belonged to class I, with a significantly higher prevalence of abdominal pain, shock, myocarditis, lymphopenia, markedly elevated CRP, ferritin, troponin, and pro-BNP. Class II consisted of 29.6% of the patients who were more likely to have respiratory complaints resembling acute COVID-19, with a higher mortality rate. The remaining 34.7% of the patients were in class III, with younger age, and presented predominantly with Kawasaki-like features [16]. In the present study, we showed that our MIS-C patients were clustered into four subgroups. Patients in cluster I were younger and presented predominantly with Kawasaki-like features. Patients in cluster II expressed predominantly MAS-like features, while patients in cluster III manifested predominantly LV dysfunction. Cluster IV was the mildest form with lower CRP, ferritin, IL-6, and procalcitonin levels. Besides the clinical and laboratory findings, outcomes were quite different in each group. Patients in clusters II and III required intensive care, pulse steroid, anakinra, and inotropic agents more frequently. Furthermore, patients in cluster IV recovered faster than others. The mortality rate was 2.3%, and of those, 57.2% were labeled as cluster II, while 42.8% were labeled as cluster III.

According to a recent meta-analysis, higher levels of WBC, absolute neutrophil counts, CRP, D-dimer, and ferritin, and lower levels of absolute lymphocyte counts were found to be associated with the severe course of MIS-C [17]. Correspondingly, in the present study, patients who experienced severe phenotype had higher levels of NLR, MPV, procalcitonin, ferritin, IL-6, fibrinogen, D-Dimer, pro-BNP, and troponin-I, and lower levels of lymphocyte and platelet counts.

Since MIS-C is an emerging phenomenon, randomized, controlled trials are insufficient to guide the treatment. Although IVIG is the first preferred one, steroids, and biological drugs such as anakinra and tocilizumab are also among the main treatment options. Steroids may be considered as a treatment option in milder forms, while severe or refractory cases should receive more intensive treatment [18,19]. Accordingly, patients in clusters II and III more commonly required intensive care, pulse steroid, anakinra, and inotropic agents.

The main limitation of the study is its national design, which does not allow to examine the influence of ethnic factors. Furthermore, the sample size was relatively low. Our provided data revealed that MIS-C is not a single disease, and the spectrum of MIS-C includes not just Kawasaki-like disease but other diseases such as MAS, LV systolic dysfunction, and a milder form as well. Since all patients do not present the same clinical features, outcomes vary in each patient; thus, understanding the disease spectrum will provide an individualized management approach.

## Figures and Tables

**Table 1 jcm-11-01742-t001:** Clinical characteristics of MIS-C patients.

	Patients with MIS-C (*n* = 293)
Fever, *n* (%)	293 (100)
Mucocutaneous features	
Polymorphous rash, *n* (%)	173 (59)
Conjunctivitis, *n* (%)	156 (53.2)
Oral changes, *n* (%)	107 (36.5)
Extremity changes, *n* (%)	49 (16.7)
Cervical lymphadenopathy, *n* (%)	77 (26.2)
Mesenteric lymphadenopathy, *n* (%)	29 (9.8)
Hepatomegaly, *n* (%)	31(10.5)
Splenomegaly, *n* (%)	19 (6.4)
Musculoskeletal features	
Myalgia, *n* (%)	80 (27.3)
Arthralgia, *n* (%)	65 (22.1)
Respiratory findings	
Cough, *n* (%)	22 (7.5)
Dyspnea, *n* (%)	22 (7.5)
Gastrointestinal findings	
Abdominal pain, *n* (%)	183 (62.4)
Nausea and vomiting, *n* (%)	114 (38.9)
Peritonitis, *n* (%)	37 (12.6)
Diarrhea, *n* (%)	102 (34.8)
Bloody diarrhea, *n* (%)	53 (18.1)
Cardiac involvement	
Hypotension, *n* (%)	90 (30.7)
Tachycardia, *n* (%)	86 (29.3)
Bradycardia, *n* (%)	10 (3.4)
LV dysfunction or myocarditis, *n* (%)	56 (19.1)
Aortic valve insufficiency, *n* (%)	14 (4.7)
Mitral valve, *n* (%)	63 (21.5)
Pericarditis, *n* (%)	27 (9.2)
Coronary artery involvement, *n* (%)	25 (8.5)
Renal involvement, *n* (%)	3 (1)
Neurologic involvement	
Headache, *n* (%)	43 (14.6)
Loss of consciousness, *n* (%)	17 (5.8)

**Table 2 jcm-11-01742-t002:** Laboratory findings of patients at admission.

Complete Blood Count **	
WBC *, mm^3^	9900 (1300–51,530)
Lymphocyte, mm^3^	1120 (160–9410)
NLR *	5.7 (0.2–72)
Hemoglobin, g/dL	11.3 (3.1–15.5)
Platelet, mm^3^	198,000 (11900–1,050,000)
MPV *, fL	9.4 (6.7–12.5)
PDW *, fL	15.8 (6.6–20.3)
PCT *, %	0.18 (0–9.08)
Inflammatory markers **	
CRP *, mg/L	40 (0.1–110)
ESR *, mm/hr	50 (2–140)
Procalcitonin, ng/mL	2.7 (0.001–100)
Ferritin, ug/L	328 (12.8–20,173)
IL-6 *, pg/mL	27 (2–2330)
Cardiac markers **	
NT-pro-BNP *, ng/L	1150 (12–70,000)
Troponin-I, ng/L	0.004 (0–89.7)
Coagulation parameters **	
D-dimer, µg/mL	2.93 (0.18–26.9)
Fibrinogen, g/L	504 (130–1096)

* CRP, C-reactive protein; ESR, erythrocyte sedimentation rate; IL-6, interleukin-6; MPV, mean platelet volume; NLR, neutrophil–lymphocyte ratio; NT-pro-BNP, N-terminal prohormone brain natriuretic peptide; PCT, platelet pressure; PDW, platelet distribution width; WBC, white blood count; ** data expressed as median (minimum–maximum).

**Table 3 jcm-11-01742-t003:** The characteristics of the patients according to clusters.

	Cluster I (*n* = 100)	Cluster II (*n* = 34)	Cluster III (*n* = 47)	Cluster IV (*n* = 112)
Age, years	60 (5–200)	126 (7–214)	120 (10–208)	97 (1–210)
Duration of fever, days	5 (1–15)	7 (1–30)	5 (1–10)	4 (1–21)
Length of hospitalization, days	8 (4–21)	11(5–39)	12 (5–34)	8 (3–32)
Gender (Male/Female)	54/46	18/16	27/20	77/35
Polymorphous rash, *n* (%)	74 (74)	20 (58.8)	29 (61.7)	50 (44.6)
Conjunctivitis, *n* (%)	78 (78)	17 (50)	32 (68.1)	29 (25.9)
Oral changes, *n* (%)	67 (67)	13 (38.2)	15 (31.9)	12 (10.7)
Extremity changes, *n* (%)	29 (29)	9 (26.5)	4 (8.5)	7 (6.2)
Cervical lymphadenopathy, *n* (%)	44 (44)	9 (26.5)	12 (25.5)	12 (10.7)
Mesenteric lymphadenopathy, *n* (%)	5 (5)	3 (8.8)	4 (8.5)	17 (15.1)
Splenomegaly, *n* (%)	4 (4)	5 (14.7)	5 (10.6)	5 (4.4)
Myalgia, *n* (%)	35 (35)	8 (23.5)	14 (29.7)	23 (20.5)
Arthralgia, *n* (%)	27 (27)	10 (29.4)	11 (23.4)	17 (15.1)
Respiratory findings	9 (9)	9 (26.5)	14 (29.7)	7 (6.2)
Gastrointestinal involvement, *n* (%)	63 (63)	24 (70.5)	30 (63.8)	80 (71.4)
Left ventricular systolic dysfunction, *n* (%)	6 (6)	2 (5.8)	47 (100)	1 (0.9)
Coronary artery involvement, *n* (%)	18 (18)	3 (8.8)	0 (0)	4 (3.5)
Renal involvement, *n* (%)	0 (0)	3 (8.8)	0 (0)	0 (0)
Neurologic involvement, *n* (%)	15 (15)	8 (23.5)	13 (27.6)	18 (16)
Requirement of intensive care, *n* (%)	11 (11)	20 (58.8)	42 (89.3)	14 (12.5)
Complete blood count **				
WBC *, mm^3^	9890 (3520–51,530)	4020 (3200–47,500)	9560 (1680–38,370)	9780 (1300–37,970)
Lymphocyte, mm^3^	1460 (228–8518)	420 (390–5540)	790 (160–4320)	1142 (200–9410)
NLR *	4.24 (0.4–20.2)	4.8 (0.2–31.6)	12.6 (2.4–28.03)	5.3 (0.4–72)
Hemoglobin, g/dL	11.1 (7.5–13.6)	9.6 (3.1–14.1)	11.5 (8.7–14.3)	11.6 (4.5–15.5)
Platelet, mm^3^	211,000 (32,000–1,050,000)	108,000 (67,500–715,000)	158,000 (65,600–369,100)	210,000 (11,900–627,000)
Inflammatory markers **				
CRP *, mg/L	14.4 (0.3–37.8)	44.2 (11.4–38.2)	18.4 (2.9–38)	11.6 (0.1–110)
ESR *, mm/h	62 (2–140)	16 (2–131)	39 (2–112)	46.5 (2–128)
Procalcitonin, ng/mL	2.09 (0.01–100)	8.1 (0.31–100)	2.7 (0.1–100)	1.2 (0.02–100)
Ferritin, ug/L	245 (35–2019)	1134 (284–20,173)	674 (44–19,878)	187 (12.8–2585)
IL–6 *, pg/mL	19.8 (2.06–1000)	56.5 (3–1000)	179 (3.7–2330)	16.5 (2–1000)
Cardiac markers **				
NT–pro–BNP *, ng/L	1150 (12–23,996)	722 (39–35000)	3878 (164–70,000)	360 (10–350,000)
Troponin–I, ng/L	0.003 (0–0.14)	0.011 (0–3.27)	0.02 (0–89.7)	0.004 (0–3.75)
Coagulation parameters **				
D-dimer, µg/mL	2.39 (0.28–22.5)	3.87 (0.64–23.3)	4.14 (0.11–26.9)	1.95 (0.18–20)
Fibrinogen, g/L	526 (245–850)	189 (157–992)	556 (133–1096)	372 (130–1061)

* CRP, C-reactive protein; ESR, erythrocyte sedimentation rate; IL-6, interleukin-6; MPV, mean platelet volume; NLR, neutrophil–lymphocyte ratio; NT-pro-BNP, N-terminal prohormone brain natriuretic peptide; PCT, platelet pressure; PDW, platelet distribution width; WBC, white blood count; ** data expressed as median (minimum–maximum).

**Table 4 jcm-11-01742-t004:** Comparison of patients with the severe course and without severe course.

	Severe Course (*n* = 81)	Without Severe Course (*n* = 212)	*p*-Value
Age, years	120 (1–214)	76 (5–210)	0.01
Duration of fever, days	5 (1–30)	4 (1–21)	0.09
Length of hospitalization, days	13 (5–39)	8 (3–27)	<0.001
Polymorphous rash, *n* (%)	53 (65.4)	120 (56.6)	0.16
Conjunctivitis, *n* (%)	48 (59.2)	108 (50.9)	0.15
Oral changes, *n* (%)	26 (32.1)	81 (38.2)	0.35
Extremity changes, *n* (%)	12 (14.8)	37 (17.4)	0.57
Cervical lymphadenopathy, *n* (%)	20 (24.6)	57 (26.8)	0.74
Mesenteric lymphadenopathy, *n* (%)	8 (9.8)	21 (9.9)	0.96
Hepatomegaly, *n* (%)	15 (18.5)	16 (7.5)	0.006
Splenomegaly, *n* (%)	11 (13.5)	8 (3.7)	0.002
Respiratory findings	11 (13.5)	11 (5.2)	0.009
Gastrointestinal involvement, *n* (%)	58 (71.6)	139 (65.5)	0.32
Renal involvement, *n* (%)	3 (3.7)	0 (0)	0.02
Neurologic involvement, *n* (%)	30 (37.1)	24 (11.3)	<0.001
Complete blood count **			
WBC, mm^3^	8095 (3550–14,440)	9100 (1300–51,530)	0.626
Lymphocyte, mm^3^	925 (160–5300)	1688 (285–9410)	0.001
NLR	6.5 (3–72)	2.4 (0.2–28.3)	<0.001
Hemoglobin, g/dL	11.3 (3.1–14.7)	11 (7.7–15.5)	0.078
Platelet, mm^3^	168,000 (65,600–692,000)	212,000 (11,900–1,050,000)	<0.001
MPV, fL	10 (6.7–12.5)	9 (6.7–12.5)	<0.001
PDW, fL	16.2 (6.6–18.39)	15.8 (7.6–20.3)	0.059
PCT, %	0.18 (0–0.35)	0.19 (0–9.08)	0.051
Inflammatory markers **			
CRP, mg/L	47 (10–110)	38 (0.1–98)	0.115
ESR, mm/h	42.5 (9–69)	55 (2–140)	0.024
Procalcitonin, ng/mL	2.8 (0–72)	2.2 (0.02–100)	0.009
Ferritin, ug/L	573 (127–20,173)	214.5 (12.8–2275)	<0.001
IL-6, pg/mL	37 (2–2330)	17.2 (0–1000)	0.027
Cardiac markers **			
NT-pro-BNP, ng/L	3500 (12–35,000)	474 (39–70,000)	<0.001
Troponin-I, ng/L	0.047 (0–89.7)	0.003 (0–37)	<0.001
Coagulation parameters **			
D-dimer, µg/mL	10.09 (2.35–26.9)	2.05 (0.18–3.86)	<0.001
Fibrinogen, g/L	620 (130–1096)	507 (240–796)	0.008

CRP, C-reactive protein; ESR, erythrocyte sedimentation rate; IL-6, interleukin-6; MPV, mean platelet volume; NLR, neutrophil–lymphocyte ratio; NT-pro-BNP, N-terminal prohormone brain natriuretic peptide; PCT, platelet pressure; PDW, platelet distribution width; WBC, white blood count; ** data expressed as median (minimum–maximum).

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
