# Peer review of "The Multifaceted Presentation of the Multisystem Inflammatory Syndrome in Children: Data from a Cluster Analysis"

_jcm, 2022, doi:10.3390/jcm11061742_

Round 1

Reviewer 1 Report

Authors present a valuable work based on numerous MIS-C groups of patients, but many data are unprecise making it less trustful and hard to compare to other reports. 

Line 92: define organ dysfunction 

Line 115: define obesity (according body weight or BMI centiles/ z-scores/ subjectively?)

Patients recruited from rheumatological wards, not from general pediatric wards - could influence sampling.

Patients with underlying rheumatological disorders should be excluded from the analysis as their picture of disease could be influenced by the chronic disease or treatment; MAS risk is increased for these patients and chronic disease exacerbation could fulfill MIS-C criteria and influence the results while not representing MIS-C cases. 

Table 1. Define hypotension, tachycardia, bradycardia, LV dysfunction, and myocarditis (were suspected, probable, confirmed cases involved?). What does mitral valve stand for - insufficiency? How were valves insufficiencies defined? How was CA involvement defined - including dilation or aneurysms only, which criteria were used to evaluate the coronary arteries?

127 What were the thresholds used to define lymphopenia or thrombocytopenia. For most of the laboratory parameters - were they increased according to laboratory thresholds (so even a slight increase would be included in the analysis) or did authors define the common threshold for all the settings involved in the study? It is especially interesting considering that the CRP median, as well as min-max values, are way lower than in current reports; a higher portion of children didn't have increased CRP at all.

144-147 the statistical comparison of clusters between waves is necessary;  the argument about different virus variants is way too bold. There were many factors that could influence it, even as simple as different vigilance and ability to recognize the disease by doctors, at the beginning, there was a connection with KD suggested, and KD-like was most commonly used to inform the HCW about the new disease. Besides that, it is not only the variants that changed but the immunological background of the population and many more factors. 

155-156 If you define a cluster as KD-like or LV-disfunction it is obvious that defining symptoms would be more often in it, so it shouldn't be compared between the clusters anymore. 

While many presented results are no surprise and at the current stage of knowledge don't reveal important information that could influence clinical decisions, the analysis of laboratory and clinical dynamics of changes is still very interesting and I believe that the manuscript would gain a lot if you worked more on this section (including treatment implementation time and characteristics) 

192 what was the death cause of these patients? 

Table 4 define the severe course

 Conclusions about treatment in discussion are not well grunted within results.

Author Response

Reviewer 1:

Comment 1: Authors present valuable work based on numerous MIS-C groups of patients, but many data are unprecise making it less trustful and hard to compare to other reports. 

Response 1: Thank you for your excellent suggestions and recommendations. We hope the revised manuscript has addressed all the concerns that you highlighted and the revised version of the manuscript is more clear.

Comment 2: Line 92: define organ dysfunction 

Response 2: We defined organ dysfunction as follows:

“……….having multiple organ dysfunction syndrome (MODS). MODS is defined as the progressive physiological dysfunction of two or more organ systems where homeostasis cannot be maintained without intervention.”

Comment 3: Line 115: define obesity (according body weight or BMI centiles/ z-scores/ subjectively?)

Response 3: We defined obesity as follows:

….” obese (age and gender-specific body mass index (BMI) ≥95th percentile).”

Comment 4: Patients recruited from rheumatological wards, not from general pediatric wards - could influence sampling.

Response 4: This is a good point. In our country, patients with MIS-C were followed up by division of infection disease, cardiology, and rheumatology, collaboratively. Therefore, all patients were hospitalized in general pediatric wards. So, there was no selective bias.

Comment 5: Patients with underlying rheumatological disorders should be excluded from the analysis as their picture of disease could be influenced by the chronic disease or treatment; MAS risk is increased for these patients and chronic disease exacerbation could fulfill MIS-C criteria and influence the results while not representing MIS-C cases. 

Response 5: This is another good point. We agreed that chronic diseases or treatments may influence the result. So we added this issue as a limitation.

“…. In present data, among MIS-C patients, 12 had a concomitant rheumatic disease that may facilitate the progress of MIS-C. ….”

Comment 6: Table 1. Define hypotension, tachycardia, bradycardia, LV dysfunction, and myocarditis (were suspected, probable, confirmed cases involved?). What does mitral valve stand for - insufficiency? How were valves insufficiencies defined? How was CA involvement defined - including dilation or aneurysms only, which criteria were used to evaluate the coronary arteries?

Response 6: We defined these terms in the method section, thank you.

“Less than the 5th percentile of age and gender-specific blood pressure levels were defined as hypotension. Heart rates were evaluated according to the normal range for age. Left ventricular dysfunction was defined as decreased EF (<55%) or FS (<28%). Coronary artery abnormalities were defined according to the AHA guidelines for KD as dilation (z-score = 2.0-2.49), small aneurysm (z-score = 2.5-4.9), moderate aneurysm (z-score = 5-9.9) and large/giant aneurysm (z-score ≥10) [8]. Mitral valve insufficiency was determined as the malfunction of the mitral valve that results in regurgitation from the left ventricle into the left atrium.”

Comment 7: 127 What were the thresholds used to define lymphopenia or thrombocytopenia. For most of the laboratory parameters - were they increased according to laboratory thresholds (so even a slight increase would be included in the analysis) or did authors define the common threshold for all the settings involved in the study? It is especially interesting considering that the CRP median, as well as min-max values, are way lower than in current reports; a higher portion of children didn't have increased CRP at all.

Response 7: We defined lymphopenia or thrombocytopenia. Elevated CRP levels were detected in 265 (86.8%). The remaining 40 patients had elevated procalcitonin levels.

Comment 8: 144-147 the statistical comparison of clusters between waves is necessary;  the argument about different virus variants is way too bold. There were many factors that could influence it, even as simple as different vigilance and ability to recognize the disease by doctors, at the beginning, there was a connection with KD suggested, and KD-like was most commonly used to inform the HCW about the new disease. Besides that, it is not only the variants that changed but the immunological background of the population and many more factors. 

Response 8: We agreed with reviewer 1. Many factors may influence the disease course. Therefore, we added as a limitation.

“……………..The main limitation of the study is its national design that does not allow to examine the influence of ethnic factors. Furthermore, we cannot underestimate the influence of different variants on disease course. Different phenotypes may originate from different variants. In present data, among MIS-C patients, 12 had a concomitant rheumatic disease that may facilitate the progress of MIS-C. ………….”

 Comment 9: 155-156 If you define a cluster as KD-like or LV-disfunction it is obvious that defining symptoms would be more often in it, so it shouldn't be compared between the clusters anymore. 

Response 9: We did not group the patients according to their symptoms. Cluster analysis is a statistical method that divided patients according to their most common features.  

Comment 10: While many presented results are no surprise and at the current stage of knowledge don't reveal important information that could influence clinical decisions, the analysis of laboratory and clinical dynamics of changes is still very interesting and I believe that the manuscript would gain a lot if you worked more on this section (including treatment implementation time and characteristics) 

Response 10: . We could not understand this comment. If you clarify, we would lite to revise.

Comment 11: 192 what was the death cause of these patients? 

Response 11: They died due to multisystem organ failure

Comment 12: Table 4 define the severe course.

Response 12: We defined this in the method section.

Comment 13: Conclusions about treatment in the discussion are not well grunted within results.

Response 13: We revised the treatment part, thank you.

Reviewer 2 Report

The authors have depicted multifaceted presentation of MIS-C. Even though there were several attempts in classifying MIS-C in to specific groups, the authors introduce a interesting and distinctive classification.

The manuscript is generally well readable and seems to be well written. However, the authors should clearly explain how they came up to a 4 cluster classification. The authors may have intuitively noticed the differences between groups. But I think a better explanation is demanded. If such groups were noticed sequentially by multiple waves of COVID-19 pandemic, the groups may have originated from different variants. So if possible, the authors should try to provide information upon variant types of each wave. Again, reasonable explanation on how the authors reached to four groups is in need.

A few points:

lines 33-34: I'm not sure whether these explanations are of value, because as the authors have grouped them to each group by their characteristics, cluster I should definitely have Kawasaki-like features. Likewise, cluster III should have LV dysfunctions. It seems to be repetition of (the) same contents.

lines 34-35: Even more, the CBC results fit into the lab results of MAS.

line 65: Kawasaki disease (CH), what is CH?

line 140: Please check the word 'decelerated'. 

lines 173-175: The authors describe each percentages of the medication used. But what about concomitant uses. Currently the concomitant use of IVIG+methylPd is recommended by several guidelines. If possible, the authors should consider providing concomitant uses of medications.

Author Response

Reviewer 1:

Comment 1: The authors have depicted a multifaceted presentation of MIS-C. Even though there were several attempts in classifying MIS-C into specific groups, the authors introduce an interesting and distinctive classification.

Response 1: Thank you for your excellent suggestions and recommendations. We hope the revised manuscript has addressed all the concerns that you highlighted.

Comment 2: The manuscript is generally well readable and seems to be well written. However, the authors should clearly explain how they came up to a 4 cluster classification. The authors may have intuitively noticed the differences between groups. But I think a better explanation is demanded. If such groups were noticed sequentially by multiple waves of the COVID-19 pandemic, the groups may have originated from different variants. So if possible, the authors should try to provide information upon variant types of each wave. Again, a reasonable explanation on how the authors reached to four groups is in need.

Response 2: Thank you for giving us the opportunity to explain this matter. Cluster analysis (CA) is a different statistical method that provides us to reveal hidden phenotypes of diseases in large data sets. The goal of CA is to distinguish subphenotypes of disease by clustering them into comparable groups. With this method, researchers may reach relatively homogeneous subgroups and may individualize management or have an opportunity to study etiopathogenesis in a more homogeneous group. In the present study, we used CA for this purpose. We clarified this issue in the text, as follows:

“…………..Cluster analysis (CA) is a different statistical method that provides us to reveal hidden phenotypes of diseases in large data sets. The goal of CA is to distinguish subphenotypes of disease by clustering them into comparable groups. With this method, researchers may reach relatively homogeneous subgroups and may individualize management or have an opportunity to study etiopathogenesis in a more homogeneous group.

Herein, we aimed to define the spectrum and phenotypic characteristics and outcomes of MIS-C patients by using CA Method and to display the prognostic factors for the severe course.”

Furthermore, we also believed that waves may change the disease course. Thus we added as a limitation as follows:

“Furthermore, we can not underestimate the influence of different variants on disease course. Different phenotypes may originate from different variants.”

Comment 3: A few points: lines 33-34: I'm not sure whether these explanations are of value because as the authors have grouped them to each group by their characteristics, cluster I should definitely have Kawasaki-like features. Likewise, cluster III should have LV dysfunctions. It seems to be the repetition of (the) same contents.

Response 3: We did not group the clusters ourselves, it was separated subgroups by using CA statistical method. Actually, you are right, we have transformed our clinical observation into concrete data. Since the beginning of the pandemic, we have observed that these patients do not follow the same course, but there is no concrete statistical analysis or study that demonstrates this. In this respect, we believe that our work is valuable. We also removed the repetitive parts.

Comment 4: lines 34-35: Even more, the CBC results fit into the lab results of MAS.

Response 4: We observed lymphopenia in 189 (61.9%) and thrombocytopenia in 83 (27.2%) patients. Not only MAS patients but also most of the patients had lymphopenia and we observed that laboratory findings revealed lower lymphocyte and platelet counts and higher acute phase reactants (APRs) in cluster II while patients in cluster IV showed less inflammation with lower APRs.

Comment 5: line 65: Kawasaki disease (CH), what is CH?

Response 5: Sorry for this mistake, we corrected it.

Comment 6: line 140: Please check the word 'decelerated'. 

Response 6: Sorry for this mistake, we corrected it.

Comment 7: lines 173-175: The authors describe each percentage of the medication used. But what about concomitant uses. Currently, the concomitant use of IVIG+methylPd is recommended by several guidelines. If possible, the authors should consider providing concomitant uses of medications.

Response 7: We revised this as follows:

“Fifty-six (18.3%) patients were additionally treated with pulse methylprednisolone (10-30 mg/kg/day, 1-3 days) and 180 (59.1%) were also treated with methylprednisolone (2 mg/kg/day).”

Reviewer 3 Report

Line 58, change reference

Line 86, check the year? May 2020 or May 2021

Line 119, 120 needs to rewritten

Table 2 gives median values, not %; check  lines 127-132

Author Response

Comment 1: Line 58, change reference

Response 1: We corrected the reference.

Comment 2: Line 86, check the year? May 2020 or May 2021

Response 2: We corrected the year.

Comment 3: Line 119, 120 needs to be rewritten

Response 3: We rewrote lines 119 and 120.

“The results of the SARS-CoV-2 antibody test were available for all the patients. Of 305 patients, 293 (96.1%) had positive results of SARS-CoV-2 IgG and/or IgM antibodies.”

Comment 4: Table 2 gives median values, not %; check  lines 127-132

Response 4: We corrected, thank you.

Round 2

Reviewer 1 Report

I appreciate the corrections included by the authors, the article gained much clarity. The choice of statistical method is not perfect - clustering should be reserved for bigger sample size and in used sampling algorithm each continuous variable is assumed to have a normal (Gaussian) distribution and each categorical variable is assumed to have a multinomial distribution. What is more, if an algorithm for clustering is used, more input should be put to verify the clinical revalence of its findings. I do appreciate that the authors translated to the readers what is cluster sampling, as most clinicians may not be familiar with it. I would also like to see that authors know how to use this tool wisely, considering its choice was correct. In the results section, it should be clarified - which variables defined the clusters and within the defined samples - what makes it clinically relevant to have this classification.  And further in the discussion - how could the knowledge about coexisting of such clusters imply our approach to the patient? 

On one hand, the authors based on the clusters, on the other - assign the cases into the mild and severe diseases by themselves. Did clustering correspond to "manual" asignmend? 

What is more, the authors did the opposite to the suggested assumption about the relevance of the virus types on disease phenotypes, the results section was not improved with the provision of detailed data - how were the waves compared? What was the proportion of clusters in the next waves? No precise data was presented. What is more, you can not argue the variability over time with virus type changes only, so again - it must be corrected. The authors don't even present the changes of SARS-CoV-2 variant domination in time in comparison to MIS-C waves and cluster predominance. Including such results, they could discuss it. But still, it wouldn't prove casualty. 

Please, remove these: "Furthermore, we cannot underestimate the influence of different
variants on disease course. Different phenotypes may originate from different variants. " - it is impossible to estimate the relevance of different variants in this study. Not to underestimate.

I am also disappointed, that patients with chronic rheumatic disorders were not excluded from the statistical analysis.  

According to the dynamics of laboratory changes comment - if the authors have day-by-day blood test results it would be interesting to see whether the normalization of most important results for MIS-C was unique within clusters, whether it corresponded to the treatment used and clinical findings. 

To sum up:

The manuscript is much better than it was, but revision would make it again more valuable.

Results should either include precise MIS-C waves comparison analysis or this section should be excluded. Not enough data is presented here, impossible to evaluate the statistical relevance of findings.  

Limitations of the study should be improved 

This is the absolute minimum needed to find the manuscript acceptable for me. To make it even better I would exclude chronic rheumatological patients from the study group and dig deeper into cluster clinical meaning. 

Author Response

Reviewer 1

Comment 1

I appreciate the corrections included by the authors, the article gained much clarity. The choice of statistical method is not perfect - clustering should be reserved for bigger sample size and in used sampling algorithm each continuous variable is assumed to have a normal (Gaussian) distribution and each categorical variable is assumed to have a multinomial distribution. What is more, if an algorithm for clustering is used, more input should be put to verify the clinical revalence of its findings. I do appreciate that the authors translated to the readers what is cluster sampling, as most clinicians may not be familiar with it. I would also like to see that authors know how to use this tool wisely, considering its choice was correct. In the results section, it should be clarified - which variables defined the clusters and within the defined samples - what makes it clinically relevant to have this classification.  And further in the discussion - how could the knowledge about coexisting of such clusters imply our approach to the patient? 

Response 1: We agreed with Reviewer 1, a higher sample size allows better analyses. We mentioned this in the limitation part. We also explained what is the cluster analysis in the manuscript as follows:

Introduction:

“Cluster analysis is a different statistical method that provides clinicians to reveal hidden phenotypes of diseases in large data sets. The goal of CA is to distinguish subphenotypes of disease by clustering them into comparable groups. With this method, researchers may reach relatively homogeneous subgroups and may individualize management or have an opportunity to study etiopathogenesis in a more homogeneous group.”

Methods:

“Cluster analysis was performed by using the Two-Step Cluster Analysis method. Initially, variables were selected such as continuous or categorical. The current age, duration of symptoms, and laboratory parameters were the continuous variables while gender and clinical manifestations were the categorical variables.”

Results:

“According to Two-Step Cluster Analysis, four distinct subgroups were identified. Cluster I; predominantly Kawasaki-like features (n=100), Cluster II; predominantly MAS-like features (n=34), Cluster III; predominantly LV dysfunction (n=47), Cluster IV; other presentations (n=112)………..”

“…..The need for intensive care, pulse steroid, anakinra, and inotropic agents were more common in both clusters II and III.”

Discussion:

“By phenotyping clusters for clarifying the disease spectrum, the management of the disease may be individualized. ……In the present study, we showed that our MIS-C patients were clustered into four subgroups. Patients in cluster I were younger and presented predominantly with Kawasaki-like features. Patients in cluster II expressed predominantly MAS-like features while patients in cluster III manifested predominantly LV dysfunction. Cluster IV was the mildest form with lower CRP, ferritin, IL-6, and procalcitonin levels. Besides the clinical and laboratory findings, outcomes were quite different in each group. Patients in clusters II and III required intensive care, pulse steroid, anakinra, and inotropic agents more frequently. Furthermore, patients in cluster IV recovered faster than others.  The mortality rate was 2.3% and of them, 57.2% were labeled as cluster II while 42.8% were labeled as cluster III.

Comment 2: On one hand, the authors based on the clusters, on the other - assign the cases into the mild and severe diseases by themselves. Did clustering correspond to "manual" assignment? 

Response 2: In the present study, a severe case was defined as the presence of a requirement for O2 support, vasoactive agents, or having multiple organ dysfunction syndrome (MODS). According to this definition, we divided patients according to whether they are severe or not. To clarify this part we revised the aim as follows:

“Herein,  the primary aim of the study is to define the spectrum and phenotypic characteristics and outcomes of MIS-C patients by using the CA method and the secondary aim is to display the prognostic factors for the severe course.”

Comment 3: What is more, the authors did the opposite to the suggested assumption about the relevance of the virus types on disease phenotypes, the results section was not improved with the provision of detailed data - how were the waves compared? What was the proportion of clusters in the next waves? No precise data was presented. What is more, you can not argue the variability over time with virus type changes only, so again - it must be corrected. The authors don't even present the changes of SARS-CoV-2 variant domination in time in comparison to MIS-C waves and cluster predominance. Including such results, they could discuss it. But still, it wouldn't prove casualty. 

Response 3In our country, we can not perform a specific test to identify which variant results in the phenotype. We just speculated this part according to data of healthy ministry. At that time the health ministry declared which variant was dominant, we accepted that the patients were infected with the aforementioned variant. Unfortunately, we could not confirm. Therefore, according to your comment, we removed this part.

Comment 4: Please, remove these: "Furthermore, we cannot underestimate the influence of different variants on disease course. Different phenotypes may originate from different variants. " - it is impossible to estimate the relevance of different variants in this study. Not to underestimate.

Response 4:  We removed this part according to your comment.

Comment 5: I am also disappointed, that patients with chronic rheumatic disorders were not excluded from the statistical analysis.  

Response 5: We excluded the patients with rheumatic diseases and we remake cluster analysis again.

Comment 6: According to the dynamics of laboratory changes comment - if the authors have day-by-day blood test results it would be interesting to see whether the normalization of most important results for MIS-C was unique within clusters, whether it corresponded to the treatment used and clinical findings. 

Response 6: We mentioned this comment in the result section as follows:

“….. the lymphocyte count was the first laboratory marker that returned to normal range. The lymphocyte count increased to over 1500 mm3 within a median of 13 days. CRP, D-dimer, and BNP levels came to normal within a median of 14, 20, and 32 days, respectively…………”

Comment 7: To sum up:

The manuscript is much better than it was, but revision would make it again more valuable.

Results should either include precise MIS-C waves comparison analysis or this section should be excluded. Not enough data is presented here, impossible to evaluate the statistical relevance of findings.  

Response 7:  We removed this part according to your comment.

Comment 8: Limitations of the study should be improved 

Response 8:   We revised the limitation part.

Comment 9: This is the absolute minimum needed to find the manuscript acceptable for me. To make it even better I would exclude chronic rheumatological patients from the study group and dig deeper into cluster clinical meaning. 

Response 9: According to your comments, we excluded the patients with rheumatic diseases and we remake cluster analysis again.

Reviewer 2 Report

The authors have markedly improved the manuscript. The raised issues have been clearly solved.

Author Response

We are very grateful for your care and efforts.